# Wide-In, Narrow-Out: Revokable Decoding for Efficient and Effective DLLMs

**Feng Hong**[1,*]   **Geng Yu**[1,*]   **Yushi Ye**[1]   **Haicheng Huang**[1]
**Huangjie Zheng**   **Ya Zhang**[2,3]   **Yanfeng Wang**[2]   **Jiangchao Yao**[1,✉]

[1]Cooperative Medianet Innovation Center, Shanghai Jiao Tong University
[2]School of Artificial Intelligence, Shanghai Jiao Tong University
[3]Institute of Artificial Intelligence for Medicine, Shanghai Jiao Tong University School of Medicine

{feng.hong, Sunarker}@sjtu.edu.cn

## Abstract

Diffusion Large Language Models (DLLMs) have emerged as a compelling alternative to Autoregressive models, designed for fast parallel generation. However, existing DLLMs are plagued by a severe quality-speed trade-off, where faster parallel decoding leads to significant performance degradation. We attribute this to the irreversibility of standard decoding in DLLMs, which is easily polarized into the wrong decoding direction along with early error context accumulation. To resolve this, we introduce Wide-In, Narrow-Out (WINO), a training-free decoding algorithm that enables *revokable decoding* in DLLMs. WINO employs a parallel draft-and-verify mechanism, aggressively drafting multiple tokens while simultaneously using the model's bidirectional context to verify and re-mask suspicious ones for refinement. Verified in open-source DLLMs like LLaDA and MMaDA, WINO is shown to decisively improve the quality-speed trade-off. For instance, on the GSM8K math benchmark, it accelerates inference by $6\times$ while improving accuracy by 2.58%; on Flickr30K captioning, it achieves a $10\times$ speedup with higher performance. More comprehensive experiments are conducted to demonstrate the superiority and provide an in-depth understanding of WINO. [1]

## 1 Introduction

Autoregressive (AR) large language models (Radford et al., 2018; 2019), such as the GPT series (OpenAI, 2022), have shown impressive performance in a ranging of language tasks. However, their foundational token-by-token generation mechanism introduces inherent limitations, including severe inference latency, susceptibility to error propagation (Stechly et al., 2023; Valmeekam et al., 2023), and challenges in maintaining global coherence (Mei et al., 2025). In response, Diffusion Large Language Models (DLLMs) have emerged as a compelling non-autoregressive alternative, architected to overcome these bottlenecks. By generating tokens simultaneously (Li et al., 2022), DLLMs theoretically enable massive inference acceleration, while their native bidirectional attention offers improved consistency. The immense potential of DLLMs has been showcased by proprietary, closed-source systems (*e.g.*, Mercury Coder (Inception Labs, 2025) and Gemini Diffusion (Google DeepMind, 2025)), which have demonstrated astonishing speeds exceeding 1,000 tokens per second, serving as a powerful proof-of-concept.

Despite this promise, the performance of open-source DLLMs has been still disappointing. One critical bottleneck is that they are caught in a severe quality-speed trade-off dilemma. Specifically, to achieve high-quality output, these models are often forced to decode slowly, generating just one token at a time, which negates their primary architectural advantage. As shown in Fig. 1, attempting to accelerate inference by generating multiple tokens in parallel invariably leads to a significant degradation in output quality (Nie et al., 2025). This stark trade-off has largely prevented the open-source DLLMs from becoming a viable, high-performance alternative to their AR counterparts.

---

[*] Equal contribution
[1] Code: https://github.com/Feng-Hong/WINO-DLLM

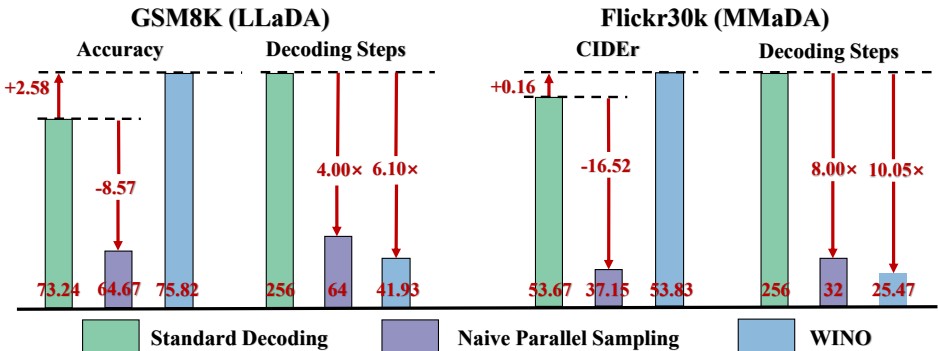

Figure 1: Demonstration of speedup and performance improvement of WINO over standard decoding and naive parallel sampling evaluated on GSM8K with LLaDA and Flickr30K with MMaDA. The standard decoding unmasks 1 token per decoding step, while the naive parallel sampling unmasks $M(>1)$ tokens per decoding step. We set $M = 4$ for GSM8K and $M = 8$ for Flickr30K.

We attribute this trade-off to a fundamental flaw in DLLMs (Sahoo et al., 2024; Ou et al., 2025): its irreversibility of the standard decoding process. Specifically, the standard generation in diffusion steps typically begins with a sequence of [MASK] tokens, which are then filled in greedily. Once a token is decoded, the decision is final and cannot be revised, even as more informative context becomes available in later steps. However, this is challenging for parallel decoding, where initial tokens are generated with very limited information, easily causing early errors to become permanently embedded, accumulated and propagated throughout the output. Therefore, such a rigid process essentially prevents DLLMs from using their greatest strength of bidirectional attention (Seo et al., 2017) to refine the early errors when the context progressively becomes rich.

To resolve this problem, we introduce Wide-In, Narrow-Out (WINO), a novel decoding algorithm that enables revokable decoding for DLLMs. WINO employs a novel draft-and-verify procedure that operates in parallel. At each step, a draft module aggressively proposes multiple new tokens based on a lenient threshold (the "Wide-In"). Concurrently, a verify module leverages the newly enriched global context to re-evaluate all previously generated tokens. Any token that fails a stricter verification check is re-masked for refinement in a future step (the "Narrow-Out"). This mechanism brings two merits: 1) it breaks the irreversibility of the conventional decoding in DLLMs, allowing the early error to be corrected for better performance; 2) it permits more aggressive token generation in each diffusion step for faster speedup with quality guarantee. Besides, our WINO is training-free and play-and-plug, which enables the general DLLMs to be both highly efficient and effective.

Our extensive experiments show that when applied to existing open-source models like LLaDA (Nie et al., 2025) and MMaDA (Yang et al., 2025), WINO achieves massive speedups, and also consistently improves model accuracy on both language and visual-language tasks. For instance, as shown in Fig. 1, on the GSM8K (Cobbe et al., 2021) math reasoning benchmark, WINO accelerates inference by 6× while simultaneously increasing accuracy by 2.58%, and on Flickr30K (Young et al., 2014) image captioning benchmark, it speedup decoding by 10× with even higher performance. By making decoding revokable, WINO unlocks the latent power of DLLMs in this area.

## 2  RELATED WORK

**Diffusion-based Language Models.** Diffusion models (Sohl-Dickstein et al., 2015; Ho et al., 2020; Song et al., 2021; Peng et al., 2026), originally popularized in image generation (Rombach et al., 2022; Nichol et al., 2022; Saharia et al., 2022), have recently gained attention as an alternative to autoregressive language models (ARLMs) for text generation. This expansion from continuous domain to discrete domain is first studied by Sohl-Dickstein et al. (2015). Subsequently, D3PM (Austin et al., 2021a) provides a general framework which models the diffusion forward process as a discrete state Markov chain defined by the multiplication of specific transition matrices over discrete time steps. Campbell et al. (2022) later expands D3PM to a continuous time setting, utilizing the theory of continuous time Markov chain(CTMC). More recently, research on masked diffusion

models(MDMs) (Shi et al., 2024) derived from the absorbing state diffusion in D3PM has shown promising results both in small-scale models (*e.g.*, MDLM (Sahoo et al., 2024) and RADD (Ou et al., 2025)) and large-scale implementations (*e.g.*, LLaDA (Nie et al., 2025) and Dream (Ye et al., 2025)). Extending this line of work, MMaDA (Yang et al., 2025) introduces a novel class of multimodal large diffusion models featuring a shared probabilistic formulation and a modality-agnostic architecture.

**DLLM Acceleration Techniques.** The existing acceleration study for DLLMs falls into two directions: KV cache and sampling compression. The former targets to build the KV cache for DLLMs due to its bidirectional full attention mechanism, unlike the causal attention of ARLMs. Typical works like Block Diffusion (Arriola et al., 2025), Fast-dLLM-cache (Wu et al., 2025) and dLLM-cache (Liu et al., 2025) respectively explore different caching mechanisms, which shows promising performance for speedup. Note that this direction is out of the scope of our work here. The latter direction focuses on optimizing the sampling process itself. For the classic low-confidence remasking strategy, several works have introduced novel sampling strategies to dynamically adjust the number of tokens predicted in parallel, thereby improving inference efficiency. Fast-dLLM-parallel (Wu et al., 2025) adopts a straightforward approach by selecting tokens with confidence scores exceeding a predefined threshold. Meanwhile, Ben-Hamu et al. (2025) propose an entropy-bounded (EB) sampler, a drop-in replacement for conventional samplers that leverages an entropy-based unmasking procedure to dynamically decode multiple tokens per step while maintaining a predefined error tolerance. Although our WINO brings the acceleration promise due to sampling compression, different from these works, we explore to address the inherent limitation of standard decoding in DLLMs. Recent follow-up work such as COVER (Xiang et al., 2026) further optimizes our revokable decoding framework by utilizing an in-place KV cache override mechanism to mitigate redundant flip-flop oscillations during verification.

## 3 PRELIMINARY: DECODING PROCESS FOR DLLMs

Given a prompt $X$, a DLLM is designed to generate a response $Y = [y_1, y_2, \ldots, y_L]$ with a predefined response length $L$. The response sequence is initialized as all special mask tokens, $Y^{(0)} = [\boxed{\text{[MASK]}}, \boxed{\text{[MASK]}}, \ldots, \boxed{\text{[MASK]}}]$. The decoding process iteratively refines the response sequence $Y^{(k)}$ over a total of $K$ denoising steps. In the following, we detail the case of $K = L$ (*i.e.*, decoding one token per step), as existing models typically achieve optimal performance under this setting (Nie et al., 2025).

At step $k$, the goal of decoding is to refine the sequence $Y^{(k-1)}$ into $Y^{(k)}$. Given the token vocabulary $V$ and the model parameterized with $\theta$, the model estimates the probability distribution over the response sequence as $p_\theta(\hat{Y}|X, Y^{(k-1)})$. As a common example, in high-confidence greedy decoding, $Y^{(k)}$ is obtained by unmasking the most confident $\boxed{\text{[MASK]}}$ token based on $Y^{(k-1)}$, *i.e.*,

$$
\begin{aligned}
l^{(k)} &= \underset{l \in \{l | y_l^{(k-1)} = \boxed{\text{[MASK]}}\}}{\arg\max} \left( \max_{v \in V} p_\theta(\hat{y}_l = v | X, Y^{(k-1)}) \right), \\
y_l^{(k)} &= \begin{cases} \underset{v \in V}{\arg\max}\, p_\theta(\hat{y}_l = v | X, Y^{(k-1)}), & \text{if } l = l^{(k)}, \\ y_l^{(k-1)}, & \text{otherwise,} \end{cases} \quad \forall l \in \{1, 2, \ldots, L\}.
\end{aligned}
\tag{1}
$$

After completing all $K$ decoding steps, the final generated response is $Y = Y^{(K)}$. Existing DLLMs, such as LLaDA (Nie et al., 2025) and MMaDA (Yang et al., 2025), can also accelerate the decoding process via naive parallel sampling by generating multiple tokens (*e.g.*, 2 or 4) per step. However, empirical results reveal that such strategies often result in substantial performance degradation, limiting their practical effectiveness despite the computational speedup (Nie et al., 2025).

**Semi-Autoregressive Diffusion Decoding.** This strategy is widely adopted by DLLMs like LLaDA (Nie et al., 2025) and MMaDA (Yang et al., 2025), which involves splitting the response sequence into multiple blocks and decoding them sequentially from left to right. Within each block, the typical diffusion decoding strategy described above is applied.

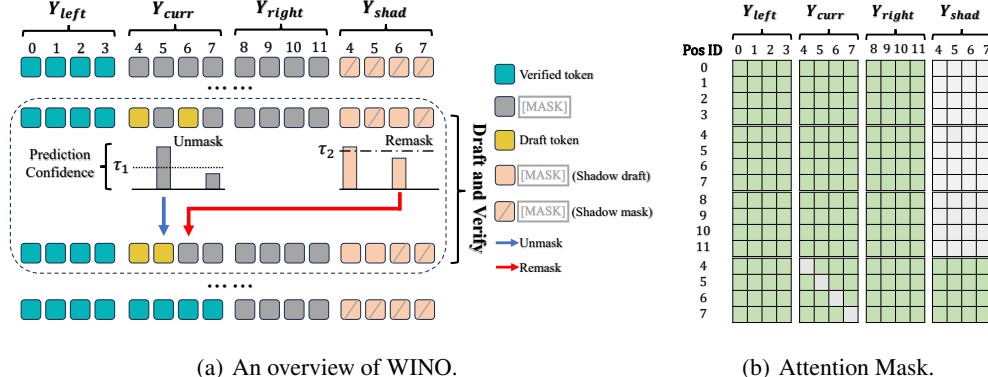

(a) An overview of WINO.

(b) Attention Mask.

Figure 2: (a) An overview of WINO. (b) Illustration of our designed attention mask. The green squares denote 1, the grey squares denote 0, and "Pos ID" is short for position ID. Verified tokens refer to tokens in the prompt $X$ or previously decoded blocks. Draft tokens denote tokens in the current block that are unmasked up to the current decoding step. [MASK] (shadow draft) refer to tokens in the shadow block whose position IDs correspond to the draft tokens while [MASK] (shadow mask) refer to the remaining tokens in the shadow block.

## 4 METHOD

### 4.1 KEY LIMITATION OF DECODING PROCESS FOR DLLMS

While architecturally suited for parallelism, DLLMs face a critical bottleneck that hinders effective multi-token decoding. During the early generation stages, the sparse context often causes the model to produce low-quality or contradictory tokens when decoding in parallel. This issue is significantly exacerbated by the standard decoding process due to its irreversibility nature. As these flawed initial predictions are permanently locked in, they inevitably propagate and degrade the final generation quality in the progressive diffusion steps, forcing an undesirable trade-off between the speed of parallel decoding and the quality of serial decoding.

To improve the trade-off, one critical point is to abandon the assumption of irreversibility in the standard decoding of DLLMs, and build a process of *revokable decoding* for progressive refinement. This principle empowers the model to iteratively refine its initial parallel outputs. As more context emerges during generation, the model can correct its preliminary predictions. Such a mechanism effectively addresses the core conflict by marrying the efficiency of parallel generation with the accuracy of context-driven corrections.

### 4.2 ITERATIVE REFINEMENT VIA PARALLEL DRAFT-AND-VERIFY

Motivated by the above analysis and the design intuition, we propose a parallel Draft-and-Verify framework to enable revokable decoding for more efficient and higher-quality generation in DLLMs.

Specifically, our framework performs two modules in parallel at each decoding step: 1) Draft: aggressively unmasks multiple [MASK] tokens into candidate meaningful tokens; 2) Verify: evaluates all currently unmasked tokens and re-masks those deemed low-quality for further refinement. We adopt the most common and general semi-autoregressive decoding paradigm to present our method. When the block length equals the generation length, it becomes equivalent to full diffusion decoding.

### 4.2.1 DRAFTING

We denote the entire sequence as $Y = [Y_{\text{left}}, Y_{\text{cur}}, Y_{\text{right}}]$, where $Y_{\text{left}}$ contains the prompt $X$ and the previously decoded blocks, $Y_{\text{cur}} = [y_{\text{cur},1}, \dots, y_{\text{cur},L_b}]$ represents the current block being decoded, and $Y_{\text{right}}$ denotes the remaining blocks to be decoded. Here, $L_b$ is the block length. At the $k$-th decoding step, instead of decoding a fixed number of tokens, we perform aggressive multi-token

parallel decoding based on a confidence threshold $\tau_1$:

$$y_{\text{cur},l}^{(k)} = \arg\max_{v \in V} p_\theta(\hat{y}_{\text{cur},l} = v|Y), \text{ if } \max_{v \in V} p_\theta(\hat{y}_{\text{cur},l} = v|Y) > \tau_1 \text{ and } y_{\text{cur},l}^{(k-1)} = \boxed{\text{[MASK]}}. \quad (2)$$

Here, a relatively low confidence threshold $\tau_1$ is adopted to allow more possible tokens to be decoded at each step, which will achieve the acceleration if only a few tokens among them are revoked during the verification module detailed in the next section. While Eq. (2) presents greedy decoding via $\mathrm{argmax}$, WINO readily supports stochastic sampling by introducing Gumbel noise to the logits ($\mathrm{Gumbel} - \mathrm{Max}$ trick) before thresholding, enabling diverse generation without altering the verification module.

### 4.2.2 VERIFICATION

The design principle of the verification module is to utilize the increasingly enriched semantic context at each decoding step—relative to earlier steps, to evaluate the quality of previously unmasked tokens. By re-masking low-quality tokens, the decoding process becomes revokable and amenable for the proper early error correction.

To realize effective quality verification about the decoded tokens, we design an auxiliary shadow block consisting entirely of $\boxed{\text{[MASK]}}$, $Y_{\text{shad}} = [\,\boxed{\text{[MASK]}}\,] \times L_b$. This block is appended to the sequence $Y$, resulting in an extended sequence $\tilde{Y} = [Y_{\text{left}}, Y_{\text{cur}}, Y_{\text{right}}, Y_{\text{shad}}]$. We carefully design the position IDs and attention mask associated with $Y_{\text{shad}}$ to ensure that its output can effectively verify the quality of the tokens decoded at the corresponding positions in $Y_{\text{cur}}$.

**Position IDs.** Although $Y_{\text{shad}}$ is appended to the right end of the sequence, we assign it the same position IDs as $Y_{\text{cur}}$. Thus, the output of $Y_{\text{shad}}$ corresponds to the same positions as $Y_{\text{cur}}$, enabling position-wise verification.

**Attention Mask.** As illustrated in Fig. 2(b), we carefully design the attention mask after incorporating $Y_{\text{shad}}$ into the sequence $\tilde{Y}$. Specifically, tokens in $Y_{\text{left}}$, $Y_{\text{cur}}$, and $Y_{\text{right}}$ can freely attend to each other, but they are not allowed to attend to $Y_{\text{shad}}$. In contrast, each token in $Y_{\text{shad}}$ is allowed to attend to all tokens except its corresponding position in $Y_{\text{cur}}$. Although indirect attention paths theoretically exist within $Y_{\text{cur}}$, our ablation study (Sec. 5.3) confirms that blocking the direct path is sufficient to prevent practical information leakage during verification.

With the above design of position IDs and attention masks, we achieve the following properties:

- For any token in the current block $Y_{\text{cur}}$, appending $Y_{\text{shad}}$ does not affect the model's output. Formally,

$$p_\theta(\hat{y}_{\text{cur},l}|Y) = p_\theta(\hat{y}_{\text{cur},l}|\tilde{Y}).$$

- For any token in $Y_{\text{shad}}$, the following properties hold. For example, consider the token $y_{\text{shad},3}$ in Fig. 2(a), which is assigned position ID 6.
    - It shares the same position ID as $y_{\text{cur},3}$, and is allowed to attend to $Y_{\text{left}}$ and $Y_{\text{right}}$;
    - It is explicitly prevented from attending to $y_{\text{cur},3}$, effectively avoiding information leakage during verification;
    - For all other positions in $Y_{\text{cur}}$, each position is attended by exactly one decoded token (from $Y_{\text{cur}}$) and one $\boxed{\text{[MASK]}}$ in $Y_{\text{shad}}$. The former provides progressively richer contextual semantics during decoding, while the latter serves to regularize the confidence of decoded tokens in $Y_{\text{cur}}$, reflecting the uncertainty and the need for potential refinement.

With the specially designed position IDs and the attention mask described above, the verification module can be formally expressed as:

$$y_{\text{cur},l}^{(k)} = \boxed{\text{[MASK]}}, \text{ if } p_\theta(\hat{y}_{\text{shad},l} = y_{\text{cur},l}^{(k-1)}|\tilde{Y})) < \tau_2 \text{ and } y_{\text{cur},l}^{(k-1)} \neq \boxed{\text{[MASK]}}, \quad (3)$$

where $\tau_2$ is the confidence threshold for verification.

### 4.2.3 Overall Procedure

In summary, at decoding step $k$, our framework enables both the drafting and verification processes to be completed in a single forward pass:

$$
y_{\text{cur},l}^{(k)} = \begin{cases} \arg\max_{v \in V} p_\theta(\hat{y}_{\text{cur},l} = v | \tilde{Y}), & \text{if } \max_{v \in V} p_\theta(\hat{y}_{\text{cur},l} = v | \tilde{Y}) > \tau_1 \text{ and } y_{\text{cur},l}^{(k-1)} = \boxed{\text{[MASK]}}, \\ \boxed{\text{[MASK]}}, & \text{if } p_\theta(\hat{y}_{\text{shad},l} = y_{\text{cur},l}^{(k-1)} | \tilde{Y})) < \tau_2 \text{ and } y_{\text{cur},l}^{(k-1)} \neq \boxed{\text{[MASK]}}, \quad (4) \\ y_{\text{cur},l}^{(k-1)}, & \text{otherwise.} \end{cases}
$$

We iteratively refine the entire $Y_{\text{cur}}$ using the procedure in Eq. (4), until all tokens in $Y_{\text{cur}}$ are no longer $\boxed{\text{[MASK]}}$. We set the drafting threshold $\tau_1$ and the verification threshold $\tau_2$ such that $\tau_1 < \tau_2$. A lower $\tau_1$ accelerates the decoding process by allowing more tokens to be generated in parallel, while a higher $\tau_2$ ensures the quality of the final output by enforcing stricter acceptance criteria. We refer to this design philosophy as *"Wide-In, Narrow-Out"* and term our method as WINO in short. To facilitate implementation and reproducibility, we provide the formal pseudocode of the WINO decoding algorithm in Sec. B.

## 5 Experiment

### 5.1 Experiment setup

**Datasets and Baselines.** We conduct experiment to evaluate WINO across different types of tasks and domains. Specifically, for language domain, we compare WINO with the standard decoding of LLaDA on eight tasks: GSM8K (Cobbe et al., 2021), MATH-500 (Hendrycks et al., 2021), HumanEval (Chen et al., 2021), MBPP (Austin et al., 2021b), Countdown (Zhao et al., 2025), Sudoku (Zhao et al., 2025), ARC-E (Clark et al., 2018), and ARC-C (Clark et al., 2018), covering four categories of textual generation tasks, including math reasoning, code generation, logical reasoning, and commonsense reasoning. For vision-language domain, we evaluate WINO against the standard decoding of MMaDA (Yang et al., 2025) on six multimodal understanding tasks: Flickr30k (Young et al., 2014), AI2D (Kembhavi et al., 2016), MATH-Vision (Wang et al., 2024), MathVista (Lu et al., 2024), MMMU (Yue et al., 2024) and ScienceQA (Lu et al., 2022), spanning four types of multimodal tasks—captioning, chart understanding, math reasoning and multi-discipline reasoning. For clarity, we test on the validation set of MMMU and the official testmini subset of MathVista.

**Evaluation Details.** All benchmarks are evaluated in a zero-shot manner, except Sudoku, which is evaluated in a 4-shot setting. We use the CIDEr metric (Vedantam et al., 2015) for the Flickr30k benchmark and accuracy for all the remaining benchmarks. To assess the inference efficiency of the decoding method, we measure the required decoding steps and Tokens Per Second (TPS) of the baselines and WINO on every task by averaging over all the samples in a benchmark. TPS is calculated as the total generated tokens divided by the total wall-clock inference time, accounting for all computational overheads (*e.g.*, shadow block computation).

**Implementation details.** We adopt the open-sourced LLaDA-8B-Instruct[2] for language benchmarks and MMaDA-8B-MixCoT[3] for vision-language tasks. We employ the semi-autoregressive sampling strategy introduced in LLaDA (Nie et al., 2025), where the output sequence is partitioned into multiple blocks and generated from left to right. In our evaluation, conducted on a single NVIDIA A100 (80GB) GPU, we set the generation length to 256 and the block length to 128, unless specified otherwise. For the hyperparameters of WINO, we set the verification threshold $\tau_2$ to 0.9 and tune the drafting threshold $\tau_1$ from $\{0.5, 0.6, 0.7\}$.

### 5.2 Main Results

**Performance and speedup on text generation.** We report the performance, decoding steps and throughput (TPS) of LLaDA, with and without WINO, on language benchmarks in Tab. 1. WINO achieves significantly better accuracy with far fewer decoding steps than the baseline LLaDA, except

---

[2]https://huggingface.co/GSAI-ML/LLaDA-8B-Instruct
[3]https://huggingface.co/Gen-Verse/MMaDA-8B-MixCoT

Table 1: Performance and inference speedup comparison on diverse language benchmarks.

| Benchmark | Method | Accuracy | Steps | Step Reduction | TPS | TPS Speedup |
|---|---|---|---|---|---|---|
| GSM8K
Math Reasoning | LLaDA | 73.24 | 256 | 1.00 × | 17.76 | 1.00 × |
| | WINO | 75.82 (+2.58) | 41.93 (-214.07) | 6.10 × | 100.53 (+82.77) | 5.66 × |
| MATH-500
Math Reasoning | LLaDA | 32.00 | 256 | 1.00 × | 17.62 | 1.00 × |
| | WINO | 34.20 (+2.20) | 74.44 (-181.56) | 3.44 × | 55.86 (+38.24) | 3.17 × |
| HumanEval
Code Generation | LLaDA | 37.80 | 256 | 1.00 × | 14.52 | 1.00 × |
| | WINO | 42.07 (+4.27) | 93.32 (-162.68) | 2.74 × | 37.19 (+22.67) | 2.56 × |
| MBPP
Code Generation | LLaDA | 36.40 | 256 | 1.00 × | 18.52 | 1.00 × |
| | WINO | 36.40 (+0.00) | 96.57 (-159.43) | 2.65 × | 45.39 (+26.87) | 2.45 × |
| Countdown
Logical Reasoning | LLaDA | 24.21 | 256 | 1.00 × | 17.22 | 1.00 × |
| | WINO | 33.20 (+8.99) | 105.88 (-150.12) | 2.41 × | 38.97 (+21.75) | 2.26 × |
| Sudoku
Logical Reasoning | LLaDA | 14.23 | 256 | 1.00 × | 11.61 | 1.00 × |
| | WINO | 15.20 (+0.97) | 131.96 (-124.04) | 1.94 × | 21.11 (+9.50) | 1.82 × |
| ARC-E
Commonsense Reasoning | LLaDA | 59.13 | 256 | 1.00 × | 17.26 | 1.00 × |
| | WINO | 81.19 (+22.06) | 40.19 (-215.81) | 6.37 × | 101.61 (+84.35) | 5.89 × |
| ARC-C
Commonsense Reasoning | LLaDA | 51.87 | 256 | 1.00 × | 17.10 | 1.00 × |
| | WINO | 73.89 (+22.02) | 47.41 (-208.59) | 5.40 × | 85.42 (+68.32) | 5.00 × |

for the MBPP task, where WINO achieves the same performance as LLaDA. For instance, WINO improves accuracy on GSM8K by 2.58% with 6.10× step reduction and 5.66× TPS speedup. Tab. 1 demonstrate the effectiveness of WINO in enhancing generation quality and inference efficiency.

**Performance and speedup on multimodal understanding and reasoning.** We assess the performance and efficiency gain of WINO incorporated into MMaDA and summarize the results in Tab. 2. Compared to the vanilla MMaDA, WINO demonstrates consistent and substantial improvements in inference efficiency across all benchmarks. Notably, the speedup effect is even more pronounced than that on textual domain tasks, when compared with results in Tab. 1. Furthermore, WINO greatly improves the task performance of MMaDA on AI2D, MMMU and ScienceQA while maintaining comparable results on Flickr30k, MATH-Vision and MathVista. These results indicate that WINO consistently delivers both performance gains and accelerated inference in the multimodal domain.

**Relation between speedup and task complexity.** As shown in Tab. 1 and Tab. 2, we observe a consistent positive correlation between the degree of speedup and task performance across all benchmarks. For instance, WINO achieves a 10.05× step reduction on the relatively simple captioning task Flickr30k, compared to only 5.73× step reduction on the more challenging math reasoning benchmark MATH-Vision. This is because models can, in principle, solve tasks they are more proficient at with lower computational cost, leaving greater room for acceleration under our decoding method. And since models are typically more confident when handling easier tasks, each decoding step in

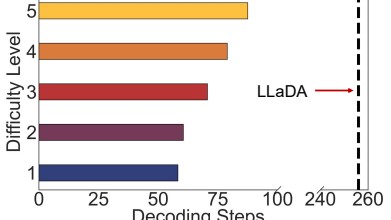

Figure 3: Decoding steps of WINO on subsets of the MATH benchmark with varied difficulty levels.

WINO tends to yield a larger number of effective tokens. To further investigate this, we evaluate the decoding steps of WINO across subsets of the MATH-500 benchmark categorized by difficulty levels. As shown in Fig. 3, WINO achieves progressively greater acceleration as the difficulty decreases, highlighting its capability to adaptively optimize inference speed based on task complexity.

### 5.3 ABLATION STUDY AND FURTHER ANALYSIS.

**On different generation length.** In Tab. 3, we evaluate the performance of WINO with experiments on different generation lengths, where the block length $L_b$ is fixed to 128 and the baselines unmask 1 token every decoding step (to achieve their best generation performance). When the generation length is set to 512, WINO still achieves comparable or better task performance with significantly fewer decoding steps, demonstrating the effectiveness of WINO across different generation lengths.

Table 2: Performance and inference speedup comparison across diverse multi-modal understanding and reasoning benchmarks. We use CIDEr for Flickr30k and accuracy for other benchmarks.

| Benchmark | Method | Performance | Steps | Step Reduction | TPS | TPS Speedup |
|---|---|---|---|---|---|---|
| Flickr30k
Captioning | MMaDA | 53.67 | 256 | 1.00 × | 6.41 | 1.00 × |
| | WINO | 53.83 (+0.16) | 25.47 (-230.53) | 10.05 × | 55.11 (+48.70) | 8.60 × |
| AI2D
Chart Understanding | MMaDA | 54.86 | 256 | 1.00 × | 6.31 | 1.00 × |
| | WINO | 57.19 (+2.33) | 30.90 (-225.10) | 8.30 × | 46.04 (+39.73) | 7.30 × |
| MATH-Vision
Math Reasoning | MMaDA | 8.55 | 256 | 1.00 × | 6.22 | 1.00 × |
| | WINO | 9.57 (+1.02) | 44.69 (-211.31) | 5.73 × | 31.17 (+24.95) | 5.01 × |
| MathVista-mini
Math Reasoning | MMaDA | 31.10 | 256 | 1.00 × | 6.21 | 1.00 × |
| | WINO | 31.40 (+0.30) | 33.45 (-222.55) | 7.65 × | 41.96 (+35.75) | 6.76 × |
| MMMU-val
Multi-discipline Reasoning | MMaDA | 18.56 | 256 | 1.00 × | 6.02 | 1.00 × |
| | WINO | 24.00 (+5.44) | 38.47 (-217.53) | 6.65 × | 36.13 (+30.11) | 6.00 × |
| ScienceQA
Multi-discipline Reasoning | MMaDA | 30.89 | 256 | 1.00 × | 6.07 | 1.00 × |
| | WINO | 42.24 (+11.35) | 28.12 (-227.88) | 9.10 × | 49.45 (+43.38) | 8.15 × |

Table 3: Experiment results on different generation lengths and full diffusion setting, respectively.

| Benchmark | Generation Length | Block Length | Method | Accuracy | Steps | Step Reduction | TPS | TPS Speedup |
|---|---|---|---|---|---|---|---|---|
| *Different Generation Lengths* | | | | | | | | |
| GSM8K | 256 | 128 | LLaDA | 73.24 | 256 | 1.00 × | 17.76 | 1.00 × |
| | | | WINO | 75.82 (+2.58) | 41.93 | 6.10 × | 100.53 | 5.66 × |
| | 512 | 128 | LLaDA | 74.60 | 512 | 1.00 × | 11.84 | 1.00 × |
| | | | WINO | 79.91 (+5.31) | 68.53 | 7.47 × | 82.64 | 6.98 × |
| MMMU-val | 256 | 128 | MMaDA | 18.56 | 256 | 1.00 × | 6.02 | 1.00 × |
| | | | WINO | 24.00 (+5.44) | 38.47 | 6.65 × | 36.13 | 6.00 × |
| | 512 | 128 | MMaDA | 18.44 | 512 | 1.00 × | 5.01 | 1.00 × |
| | | | WINO | 23.44 (+5.00) | 64.82 | 7.90 × | 35.01 | 6.99 × |
| *Full Diffusion* | | | | | | | | |
| GSM8K | 256 | 256 | LLaDA | 34.34 | 256 | 1.00 × | 17.73 | 1.00 × |
| | | | WINO | 58.22 (+23.88) | 38.77 | 6.60 × | 93.61 | 5.28 × |
| | 128 | 128 | LLaDA | 58.60 | 128 | 1.00 × | 23.23 | 1.00 × |
| | | | WINO | 62.32 (+3.72) | 23.95 | 5.34 × | 114.29 | 4.92 × |
| MMMU-val | 256 | 256 | MMaDA | 17.22 | 256 | 1.00× | 6.11 | 1.00× |
| | | | WINO | 22.44 (+5.22) | 24.94 | 10.26× | 50.03 | 8.19× |
| | 128 | 128 | MMaDA | 15.33 | 128 | 1.00× | 6.70 | 1.00 × |
| | | | WINO | 23.11 (+7.78) | 19.14 | 6.69× | 39.94 | 5.96 × |

**On full diffusion decoding (instead of semi-autoregressive decoding).** In Tab. 3, we compare the baselines and WINO applying full diffusion decoding, which means the block length $L_b$ is set equal to the generation length. Compared to results on the semi-autoregressive decoding in Tab. 1 and Tab. 2, WINO demonstrates substantially strong accuracy gains under the full diffusion setting. Notably, while LLaDA suffers a substantial accuracy drop on GSM8K with full diffusion decoding, WINO maintains reasonable performance with far fewer decoding steps. These results indicate that WINO unlocks significantly greater potential for boosting model performance and computational efficiency when applied in full diffusion decoding scenarios.

**Comparison with Other Parallel Sampling Strategies.** While naive parallel sampling accelerates decoding, it causes severe performance drops (*e.g.*, GSM8K accuracy falls to 64.67% with 4 tokens/step). We further compared WINO with advanced dynamic samplers, including Fast-dLLM-parallel (Wu et al., 2025) and Entropy-Bounded sampler (Ben-Hamu et al., 2025), which draft tokens aggressively based on confidence or entropy without revocation. As detailed in Sec. C, while these methods improve speed (approx. 3x), they still suffer from accuracy degradation (*e.g.*, Fast-dLLM-parallel achieves 72.33% on GSM8K). In contrast, WINO achieves a 5.66× speedup while boosting accuracy to 75.82%, demonstrating that the revocation mechanism is essential for breaking the quality-speed trade-off.

**Ablation on Verification and Leakage.** We validate the necessity of our verification mechanism and attention mask design in Tab. 4. First, removing verification ("Only Draft") leads to signifi-

Table 4: Experiment results on the variant of WINO without the verification module.

| Benchmark | Method | Accuracy | Steps | Step Reduction | TPS | TPS Speedup |
|---|---|---|---|---|---|---|
| GSM8K | LLaDA | 73.24 | 256 | 1.00 × | 17.76 | 1.00 × |
| | Only Draft ($\tau_1 = 0.6$) | 70.28 | 34.79 | 7.36 × | 130.89 | 7.37 × |
| | Only Draft ($\tau_1 = 0.9$) | 72.33 | 81.39 | 3.15 × | 56.12 | 3.16 × |
| | WINO (w/ Full Leakage) | 72.25 | 44.85 | 5.71 × | 92.38 | 5.20 × |
| | WINO | 75.82 | 41.93 | 6.10 × | 100.53 | 5.66 × |
| MMMU-val | MMaDA | 18.56 | 256 | 1.00 × | 6.02 | 1.00 × |
| | Only Draft ($\tau_1 = 0.6$) | 19.89 | 35.63 | 7.18 × | 43.22 | 7.18 × |
| | Only Draft ($\tau_1 = 0.9$) | 18.56 | 79.74 | 3.21 × | 19.38 | 3.22 × |
| | WINO (w/ Full Leakage) | 18.22 | 44.43 | 5.76 × | 31.54 | 5.24× |
| | WINO | 24.00 | 38.47 | 6.65 × | 36.13 | 6.00 × |

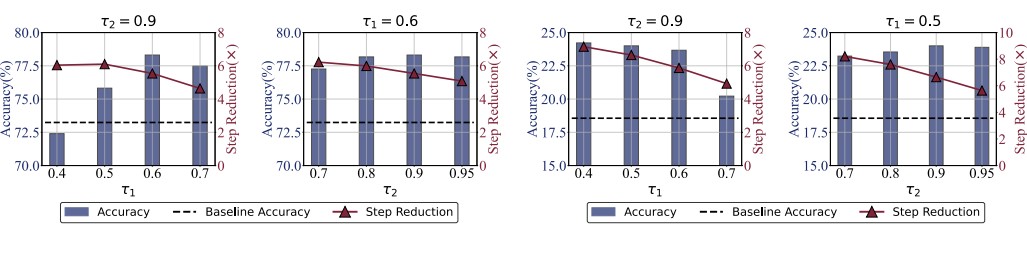

(a) WINO (LLaDA-based) on GSM8K.     (b) WINO (MMaDA-based) on MMMU-val.

Figure 4: Ablation study on the drafting threshold $\tau_1$ and the verification threshold $\tau_2$.

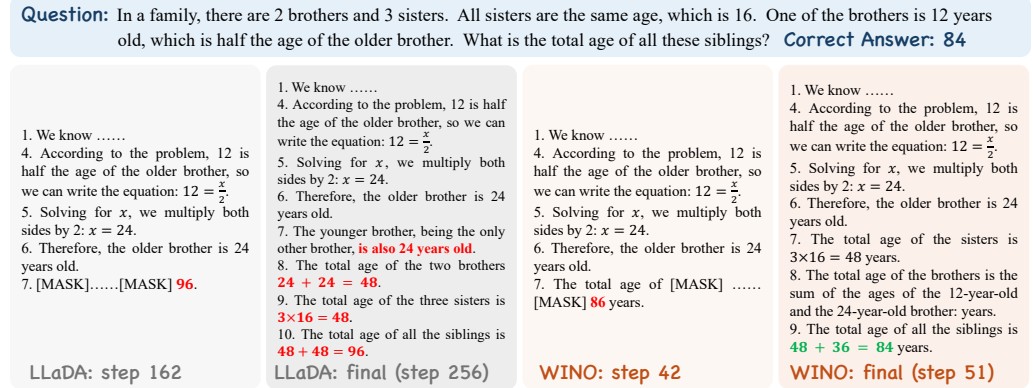

Figure 5: Case Study: GSM8K Example. We compare standard decoding with LLaDA against the intermediate and final results produced by WINO. More detailed case studies are provided in Sec. A.

cant performance degradation when using aggressive thresholds ($\tau_1 = 0.6$), as the model lacks a mechanism to correct inevitable draft errors. This variant effectively represents a dynamic sampler that aggressively drafts tokens without revocation. Second, we address the concern of information leakage. Theoretically, $Y_{\text{shad}}$ could access $Y_{\text{cur}}$ information indirectly via self-attention within $Y_{\text{cur}}$. To test if this compromises verification, we introduce "WINO (w/ Full Leakage)", where $Y_{\text{shad}}$ is allowed to *directly* attend to its corresponding token in $Y_{cur}$. As shown in Tab. 4, allowing direct leakage drops accuracy significantly (*e.g.*, -3.57% on GSM8K). This confirms that the direct attention path is the dominant factor in credit assignment. By blocking this path, our proposed mask effectively forces the model to verify tokens based on context rather than trivial identity mapping, making the potential indirect leakage negligible in practice.

**Effect of threshold tuning.** In Fig. 4, we present the evaluation results of WINO with varying drafting threshold $\tau_1$ and verification threshold $\tau_2$. Our experiments suggest that WINO consistently outperforms baselines across different benchmarks and the $\tau_1$ and $\tau_2$ values in terms of both

task performance and inference efficiency. As the $\tau_1$ value decreases, more candidate tokens are unmasked at each decoding step, thereby accelerating inference by reducing the required decoding steps. However, this comes at the cost of introducing more unreliable predictions, which may place a greater burden on the verification module to correct errors. Empirically, we find that setting the value of drafting threshold $\tau_1$ within the range of 0.5 to 0.7 achieves an optimal balance, maintaining competitive task performance while preserving efficient generation. The verification threshold $\tau_2$ controls the strictness of the verification process and thus influences decoding speed. Since the performance is relatively robust to $\tau_2$, we fix $\tau_2 = 0.9$ in all experiments, while leaving open the possibility of further tuning this parameter for even better performance and speedup.

While the main experiments in this paper focus on deterministic greedy decoding, WINO is not restricted to deterministic inference and can be naturally extended to stochastic sampling for generating diverse outputs. Specifically, during the drafting phase, we incorporate the Gumbel-Max trick to enable sampling-based token selection under a non-zero temperature.

Table 5: Performance comparison using stochastic sampling (temperature = 0.5). We report the mean and standard deviation over 3 runs.

| Benchmark | Method | Accuracy (%) | Step Reduction ($\times$) |
|---|---|---|---|
| GSM8K | LLaDA | $73.01 \pm 0.21$ | 1 |
| | WINO | $\mathbf{76.06 \pm 0.16}$ | $\mathbf{6.05 \pm 0.06}$ |
| MMMU-val | MMaDA | $18.32 \pm 0.43$ | 1 |
| | WINO | $\mathbf{23.89 \pm 0.29}$ | $\mathbf{6.69 \pm 0.09}$ |

As reported in Tab. 5, even under stochastic decoding with temperature set to 0.5, WINO consistently outperforms its corresponding baselines across both benchmarks. In particular, WINO achieves noticeable accuracy improvements while simultaneously delivering substantial inference speedups, as reflected by the step reduction factors.

**GPU memory usage.** WINO introduces an auxiliary shadow block of size $L_b$, which theoretically scales the memory overhead linearly with the block length. However, in practice, this cost remains minimal. In the standard semi-autoregressive setting ($L_b = 128, L = 256$), WINO increases usage by only $\sim 2.4\%$ (16.18GB vs. 16.57GB on GSM8K, see Fig. 6). To assess the upper bound, we further evaluated the most memory-intensive full diffusion setting ($L_b = L = 512$), where the shadow block effectively doubles the processed sequence length. Even in this extreme case, the overhead remains below $8\%$ (rising from 20.96GB to 22.65GB on MMaDA), demonstrating WINO's efficiency for longer contexts.

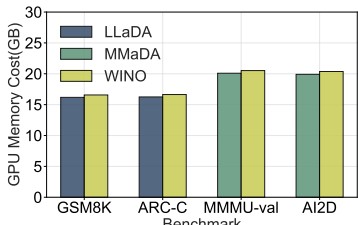

Figure 6: GPU memory usage.

**Case Study: Decoding Dynamics.** To conduct a fine-grained examination of the decoding processes of WINO, we present an example from GSM8K in Fig. 5. As shown, the baseline may produce erroneous tokens at the early decoding stages. Since the generated tokens by the baseline remain unchanged in subsequent decoding steps, the false contextual information propagates throughout the whole generation process, eventually leading to low-quality generation results. In contrast, WINO enables dynamical refinement of generated tokens via an iterative draft-and-verify mechanism, which mitigates error accumulation and facilitates high-quality decoded outputs.

# 6 CONCLUSION

In this work, we introduce Wide-In, Narrow-Out (WINO), a training-free decoding algorithm that resolves the critical quality-speed trade-off in Diffusion Large Language Models (DLLMs) by making their generation process revokable. WINO overcomes the limitations of irrevokable standard decoding by employing a parallel draft-and-verify mechanism, allowing the model to aggressively generate tokens and iteratively correct errors using its full bidirectional context. Our experiments on existing open-source models like LLaDA and MMaDA demonstrate that WINO simultaneously accelerates inference by up to $10\times$ while significantly improving accuracy across a diverse set of language and vision-language tasks. While acknowledging areas for future architectural improvements, WINO fundamentally enhances the practicality of DLLMs by rethinking the decoding process itself, establishing them as a truly efficient and high-quality alternative to autoregressive systems.

ETHICS STATEMENT

WINO is a training-free decoding algorithm that improves the efficiency and output quality of existing Diffusion Large Language Models (DLLMs). Our method operates on pre-trained models and does not involve new training data or modifications to the models' underlying parameters. Therefore, WINO does not introduce new ethical concerns beyond those already associated with the foundation models it is applied to. Any potential for bias or malicious use is a characteristic of the base model, not the decoding process we have developed.

ACKNOWLEDGEMENT

This work is supported by National Natural Science Foundation of China (No. 62306178), STCSM (No. 22DZ2229005), and the 111 Plan (No. BP0719010).

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

## USE OF LARGE LANGUAGE MODELS (LLMs)

Large language models (LLMs) (*e.g.*, GPT-5 and Gemini 2.5 Pro) were employed as an editing tool during the final preparation of this manuscript. The model's use was confined to language polishing, grammar correction, and formatting. It played no part in the substantive scientific work; all research concepts, methodologies, analyses, and conclusions were generated exclusively by the authors.

## A   CASE STUDY

**Case Study: Decoding Dynamics.** To conduct a fine-grained examination of the decoding processes of the baselines and WINO, we present two generation examples—one from GSM8K and one from MMMU-val below. As shown, the baseline may produce erroneous tokens at the early decoding stages. Since the generated tokens by the baseline remain unchanged in subsequent decoding steps, the false contextual information propagates throughout the whole generation process, eventually leading to low-quality generation results. In contrast, WINO enables dynamical refinement of generated tokens via an iterative draft-and-verify mechanism, which mitigates error accumulation and facilitates high-quality decoded outputs.

---

**Case: GSM8K, Geneation Length 256, Block Length 128, 2 blocks in total**

**Question:** In a family, there are 2 brothers and 3 sisters. All sisters are the same age, which is 16. One of the brothers is 12 years old, which is half the age of the older brother. What is the total age of all these siblings?

**GT:** 84.

---

**LLaDA:**

```
--------------------------------------------------------------------------------------------
```
**Step: 128 (1st block decoding completed). 1st block:**
```
<reasoning>
1.  We know that all three sisters are 16 years old.
2.  One of the brothers is 12 years old, which is half the age of the older brother.
3.  Let's denote the age of the older brother as x.
4.  According to the problem, 12 is half the age of the older brother, so we can write
```
the equation: $12 = \frac{x}{2}$.
```
5.  Solving for x, we multiply both sides by 2:  x = 24.
6.  Therefore, the
```
```
--------------------------------------------------------------------------------------------
```
**Step: 162 (early overconfident error). 2nd block:**
```
older brother is 24 years old.
7.<|mdm_mask|>......<|mdm_mask|>
</reasoning>
<answer>
```
96
```
</answer><|eot_id|><|endoftext|>
```
```
--------------------------------------------------------------------------------------------
```
**Final (error accumulation):**
```
<reasoning>
1.  We know that all three sisters are 16 years old.
2.  One of the brothers is 12 years old, which is half the age of the older brother.
3.  Let's denote the age of the older brother as x.
4.  According to the problem, 12 is half the age of the older brother, so we can write
```
the equation: $12 = \frac{x}{2}$.
```
5.  Solving for x, we multiply both sides by 2:  x = 24.
6.  Therefore, the older brother is 24 years old.
7.  The younger brother, being the only other brother, is also 24 years old.
8.  The total age of the two brothers is 24 + 24 = 48.
9.  The total age of the three sisters is 3 × 16 = 48.
10.  The total age of all the siblings is 48 + 48 = 96.
</reasoning>
<answer>
```
96
```
</answer><|eot_id|><|endoftext|>
```

---

**WINO:**
```
--------------------------------------------------------------------------------------------
```
**Step: 35 (1st block decoding completed). 1st block:**

```
<reasoning>
1.  We know that all three sisters are 16 years old.
2.  One of the brothers is 12 years old, which is half the age of the older brother.
3.  Let's denote the age of the older brother as x.
4.  According to the problem, 12 is half the age of the older brother, so we can write
the equation:  12 = x/2.
5.  Solving for x, we multiply both sides by 2:  x = 24.
6.  Therefore, the
```
---------------------------------------------------------------------------------------------
**Step: 37 (draft error). 2nd block:**
```
older brother is 24 years old.
7.<|mdm_mask|>......<|mdm_mask|> = 96<|mdm_mask|><|mdm_mask|><|mdm_mask|>
</reasoning<|mdm_mask|>
<answer>
```
9<|mdm_mask|>
```
</answer><|eot_id|><|endoftext|>
```
---------------------------------------------------------------------------------------------
**Step: 42 (refinement: still wrong). 2nd block:**
```
older brother is 24 years old.
7.  The total age of<|mdm_mask|> sisters<|mdm_mask|> \( 3<|mdm_mask|>times 16
= 4<|mdm_mask|>......<|mdm_mask|> total age of all the<|mdm_mask|> is \( 48 +
<|mdm_mask|><|mdm_mask|> = 86 \) years.
</reasoning>
<answer>
```
86
```
</answer><|eot_id|><|endoftext|>
```
---------------------------------------------------------------------------------------------
**Final (all correct through iterative refinement):**
```
<reasoning>
1.  We know that all three sisters are 16 years old.
2.  One of the brothers is 12 years old, which is half the age of the older brother.
3.  Let's denote the age of the older brother as x.
4.  According to the problem, 12 is half the age of the older brother, so we can write
the equation:  12 = x/2.
5.  Solving for x, we multiply both sides by 2:  x = 24.
6.  Therefore, the older brother is 24 years old.
7.  The total age of the sisters is 3 × 16 = 48 years.
8.  The total age of the brothers is the sum of the ages of the 12-year-old and the
24-year-old brother:  12 + 24 = 36 years.
9.  The total age of all the siblings is 48 + 36 = 84 years.
</reasoning>
<answer>
```
84
```
</answer><|eot_id|><|endoftext|>
```

---

**Case: MMMU-val, Geneation Length 256, Block Length 128, 2 blocks in total**

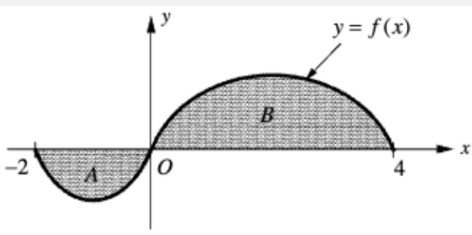

**Question:** The graph of $y = f(x)$ is shown in the figure above. The shaded region $A$ has area $a$ and the shaded region $B$ has area $b$. If $g(x) = f(x) + 3$, what is the average value of g on the interval $[-2, 4]$?
A. $(a + b + 3)/6$
B. $(-a + b + 3)/6$
C. $(-a + b)/6 + 3$
D. $(a + b)/6 + 3$

**GT: C.**

---

**MMaDA:**

```
-------------------------------------------------------------------------------------
Step: 70 (early overconfident error). 1st block:
<think>
1.  To find the average value of g(x) on the interval [−2, 4], we need to calculate the
area of the graph of g(x) over this interval.
2.  The area of the graph of g(x) is given by the sum of the areas of the two shaded
regions.
<|mdm_mask|>......<|mdm_mask|>
-------------------------------------------------------------------------------------
Step: 128 (error accumulation). 1st block:
<think>
1.  To find the average value of g(x) on the interval [−2, 4], we need to calculate the
area of the graph of g(x) over this interval.
2.  The area of the graph of g(x) is given by the sum of the areas of the two shaded
regions.
3.  The area of region A is a and the area of region B is b.
4.  Therefore, the total area of g(x) is a + b.
5.  The average value of g(x) the interval
-------------------------------------------------------------------------------------
Final (error accumulation):
<think>
1.  To find the average value of g(x) on the interval [−2, 4], we need to calculate the
area of the graph of g(x) over this interval.
2.  The area of the graph of g(x) is given by the sum of the areas of the two shaded
regions.
3.  The area of region A is a and the area of region B is b.
4.  Therefore, the total area of g(x) is a + b.
5.  The average value of g(x) on the interval [−2, 4] is (a+b)/6 .
</think>
A<|endoftext|>
```

**WINO:**

```
-------------------------------------------------------------------------------------
Step: 13 (draft error). 1st block:
<think>
1.  To find the average value of g(x) on the interval [−2, 4], we need to use
the<|mdm_mask|> for the<|mdm_mask|> of f(x) over<|mdm_mask|> interval.
2.  The average value of <|mdm_mask|>(x) is given by \frac{1}{2<|mdm_mask|> \times
\<|mdm_mask|><|mdm_mask|>2}^{4<|mdm_mask|>......<|mdm_mask|>
-------------------------------------------------------------------------------------
Step: 33 (refinement: correct). 1st block:
<think>
1.  To find the average value of g(x) on the interval [−2, 4], we need to use the formula
for the average of f(x) over an interval.
2.  The average value of f(x) is given by (1/6) · ∫_{−2}^{4} f(x) dx.
3.  Given that g(x) = f(x) + 3, we need to find f(x). The average value of f(x) is (−a+b)/6 .
4.  Therefore, <|mdm_mask|>......<|mdm_mask|>
-------------------------------------------------------------------------------------
Final (all correct through iterative refinement):
<think>
1.  To find the average value of g(x) on the interval [−2, 4], we need to use the formula
for the average of f(x) over an interval.
2.  The average value of f(x) is given by (1/6) · ∫_{−2}^{4} f(x) dx.
3.  Given that g(x) = f(x) + 3, we need to find f(x). The average value of f(x) is (−a+b)/6 .
4.  Therefore, the average value of g(x) on the interval [−2, 4] is (−a+b)/6 + 3.
5.  The correct answer is option C.
</think>
C<|endoftext|>
```

# B   WINO DECODING ALGORITHM

We present the formal algorithm for the proposed Wide-In, Narrow-Out (WINO) decoding in Algorithm 1. This algorithm details the iterative draft-and-verify loop within a single diffusion block.

# C   COMPARISON WITH ADVANCED DYNAMIC SAMPLERS

To further demonstrate that the performance gains of WINO stem from the revokable mechanism rather than merely aggressive dynamic sampling, we compared WINO against state-of-the-art dy-

---

**Algorithm 1** WINO Decoding for a Single Block

---

**Require:** Prompt context $Y_{\text{left}}$, Current block length $L_b$, Drafting threshold $\tau_1$, Verification threshold $\tau_2$, Model $p_\theta$
1: Initialize current block $Y_{\text{cur}} \leftarrow [\text{MASK}]^{L_b}$
2: Initialize shadow block $Y_{\text{shad}} \leftarrow [\text{MASK}]^{L_b}$
3: Construct extended sequence $\tilde{Y} \leftarrow [Y_{\text{left}}, Y_{\text{cur}}, Y_{\text{right}}, Y_{\text{shad}}]$
4: **while** $Y_{\text{cur}}$ contains [MASK] **do**
5:     Forward pass to get logits for $\tilde{Y}$
6:     **for** $l = 1$ to $L_b$ **do**
7:         **// Parallel Draft (Wide-In)**
8:         **if** $y_{cur,l}$ is [MASK] **then**
9:             Get candidate token $v = \arg\max_{v'} p_\theta(\hat{y}_{cur,l} = v'|\tilde{Y})$
10:           Get confidence $c_{draft} = \max_{v'} p_\theta(\hat{y}_{cur,l} = v'|\tilde{Y})$
11:           **if** $c_{draft} > \tau_1$ **then**
12:              $y_{cur,l} \leftarrow v$ {Unmask token}
13:           **end if**
14:         **else**
15:           **// Parallel Verify (Narrow-Out)**
16:           Get verification confidence $c_{verify} = p_\theta(\hat{y}_{shad,l} = y_{cur,l}|\tilde{Y})$
17:           **if** $c_{verify} < \tau_2$ **then**
18:              $y_{cur,l} \leftarrow [\text{MASK}]$ {Re-mask (Revoke)}
19:           **end if**
20:         **end if**
21:     **end for**
22:     Update $\tilde{Y}$ with new $Y_{\text{cur}}$
23: **end while**
24: **return** $Y_{\text{cur}}$

---

namic samplers: Fast-dLLM-parallel (Wu et al., 2025) and Entropy-Bounded (EB) sampler (Ben-Hamu et al., 2025). As shown in Tab. 6, while these methods achieve speedups over the baseline, they suffer from accuracy degradation (*e.g.*, Fast-dLLM drops 3.9% on GSM8K compared to WINO). WINO, utilizing the proposed verification and revocation mechanism, achieves superior speedups (approx. $6\times$) while significantly improving accuracy.

Table 6: Comparison with advanced dynamic samplers (Fast-dLLM-parallel and EB Sampler) on GSM8K and MMMU-val. WINO achieves the best trade-off between quality and speed.

| Method | Accuracy | Steps | Step Red. | TPS | Speedup |
|---|---|---|---|---|---|
| **GSM8K** | | | | | |
| LLaDA (Baseline) | 73.24 | 256.00 | $1.00\times$ | 17.76 | $1.00\times$ |
| Fast-dLLM-parallel | 72.33 | 81.39 | $3.15\times$ | 56.12 | $3.16\times$ |
| EB Sampler | 70.37 | 95.74 | $2.76\times$ | 46.71 | $2.63\times$ |
| **WINO (Ours)** | **75.82** | **41.93** | **$6.10\times$** | **100.53** | **$5.66\times$** |
| **MMMU-val** | | | | | |
| MMaDA (Baseline) | 18.56 | 256.00 | $1.00\times$ | 6.02 | $1.00\times$ |
| Fast-dLLM-parallel | 19.89 | 35.63 | $7.18\times$ | 43.22 | $7.18\times$ |
| EB Sampler | 17.87 | 61.04 | $4.20\times$ | 24.89 | $4.13\times$ |
| **WINO (Ours)** | **24.00** | **38.47** | **$6.65\times$** | **36.13** | **$6.00\times$** |

## D   Latency Ablation Study

To verify the per-step cost of WINO's custom attention mask, we measure the wall-clock time for a single forward pass on LLaDA-8B. As shown in Tab. 7, when controlling for the total sequence length (Prompt + Gen + Shadow), the WINO custom mask incurs no additional latency compared to standard full attention. This confirms that the overhead in WINO comes solely from the increased sequence length of the shadow block, not the masking mechanism itself.

Table 7: Per-step wall-clock latency ablation (ms). "Total Length" indicates the actual sequence length processed by the GPU.

| Config | Gen Len | Total Len | Attn. Type | Latency (ms) |
|---|---|---|---|---|
| Standard | 512 | 1024 | Full | $121.36 \pm 0.37$ |
| Standard (Pad) | 512 | 1152 | Full | $132.65 \pm 0.60$ |
| **WINO** | 512 | 1152 | **Custom Mask** | **$132.49 \pm 0.89$** |

# E LIMITATIONS AND FUTURE DIRECTIONS

Our work establishes WINO as a training-free framework that makes the decoding process of Diffusion Large Language Models (DLLMs) revokable, effectively addressing their quality-speed trade-off. The degree of acceleration, however, is inherently linked to the base model's capabilities; as our experiments show, more proficient models produce better drafts that require fewer refinement steps, leading to greater speedups. This insight points to a promising future direction: integrating the concept of revokable sampling directly into the training phase. A model trained with an awareness of a draft-and-verify mechanism could learn to generate more robust initial predictions and self-correct more efficiently, potentially unlocking even greater gains in performance and speed.

