# OpenReview forum: "Wide-In, Narrow-Out: Revokable Decoding for Efficient and Effective DLLMs"
_ICLR.cc/2026/Conference — ICLR 2026 Poster_

### Official Review · Reviewer_Rjq2 · 2025-10-31

**Soundness:** 3
**Presentation:** 3
**Contribution:** 2
**Rating:** 2
**Confidence:** 3

**Summary:**

Identified problem: DLLMs have to deal with quality speed trade off and it is attributed to irreversiblity of standard DLLM decoding.

The paper introduces WINO (wide in, narrow out), a training free algorithm that enables revocable decoding.

WINO's parallel draft-and verify strategy includes:
- Drafting (Wide in): aggressively draft multiple new tokens based on a lenient confidence threshold.
- Verification (Narrow out): A verification module that re-evalutes all unmasked tokens with a bit stricter verification check, remasking the ones that fail the check.

Results show good efficiency and quality gains on GSM8k and Flickr30k.

**Strengths:**

- Revokable Decoding: The paper’s main idea is to make the decoding process "revokable." This means the model can undo an early mistake and fix it right away, instead of being stuck with it (which is what happens in standard DLLMs).

- No Extra Training Needed: The WINO method is a simple algorithm change, meaning it works immediately with existing DLLMs like LLaDA and MMaDA without having to retrain them.

Improved Speed and Quality: WINO solves the trade-off between speed and accuracy. It makes the models much faster while also making them more accurate. The method provides huge gains in real-world tasks: GSM8K and Flickr30K.

**Weaknesses:**

- Please include some comparison on latency instead of decoding steps.
- Additionally add a comparison of the number of parameters in the baseline model vs combined number of params in the drafter and verifier.
- It is a bit unclear, but IIUC, the baselines do not use remasking based on low confidence, which is pretty standard and also mentioned in the related work section. Do your baselines use remasking?

**Questions:**

see weakness section

---

> ### Author Response · Authors · 2025-11-21
> **Response to Reviewer Rjq2**
>
> We appreciate your review comments. It seems that some misunderstandings may have occurred, and we would like to clarify them point by point below:
>
>
> **1. On “low-confidence re-masking” versus our Revokable Decoding**
>
> You are correct that "low-confidence re-masking" is a standard technique, and our baseline (the “standard decoding” from LLaDA) **is indeed a form of this**.
>
> However, this standard re-masking and our revokable decoding are conceptually different. Here is the distinction:
> - **Baseline (“Standard low-confidence Re-masking”)**: In this paradigm (e.g., LLaDA), at each decoding step, the model predicts probabilities for all remaining [MASK] tokens. It then selects the top M most confident tokens to un-mask (where M = 1 for the high-quality, 256-step baseline). The other, less confident tokens are "re-masked", meaning they are deferred to a future step.
>     - Key Point: Once a token is un-masked (and not re-masked) at step k, that decision is **final and irreversible**. It cannot be changed. The “re-masking” only affects the scheduling of which token to generate next.
> - **WINO ("Revokable Decoding")**: Our method directly addresses this irreversibility. WINO’s verification module evaluates tokens that were already un-masked in previous steps (e.g., at steps 1 to k−1). If a token now appears low-quality in light of the richer context, WINO revokes the prior decision and turns that token back into a [MASK].
>
> In short: The baseline’s “re-masking” is an irreversible selection process (choosing which mask to fill next), whereas WINO introduces a revokable correction process (fixing a token that was already filled).
>
> **2. On Latency**
>
> We would like to clarify that our **TPS (Tokens Per Second)** metric is, in fact, a direct measure of latency, simply presented as throughput.
>
> The **TPS Speedup** values reported in our tables are identical to the total latency reduction.
>
> TPS is calculated as $\text{Total Generated Tokens} / \text{Total Latency (in seconds)}$. Since the total number of generated tokens is held constant for both the baseline and WINO (e.g., 256 tokens for the main experiments), a 5.66x TPS speedup  directly corresponds to a 5.66x reduction in end-to-end latency.
>
> To make this more explicit, we are happy to add the following table, which converts our TPS figures from Tables 1 and 2 into absolute end-to-end latency for generating 256 tokens.
>
> | Benchmark     | Method | Latency (256 Tokens) | Latency Speedup |
> |:------------- |:------ |:-------------------- |:--------------- |
> | **GSM8K**     | LLaDA  | 14.41 s              | 1.00x           |
> |               | WINO   | **2.55 s**           | **5.65x**       |
> | **Flickr30k** | MMaDA  | 39.94 s              | 1.00x           |
> |               | WINO   | **4.64 s**           | **8.61x**       |
> | **MMMU-val**  | MMaDA  | 42.52 s              | 1.00x           |
> |               | WINO   | **7.09 s**           | **5.99x**       |
>
> This table clearly illustrates the practical latency reductions achieved by WINO. We have added this clarification to the revised manuscript (Sec 5.1).
>
> **3. On Parameters**
>
> Thank you for the question. This is a critical point of clarification.
>
> WINO is a **training-free decoding algorithm**, meaning it introduces **zero new parameters**.
>
> The "drafter" and "verifier" are not separate models. They are conceptual names for two processes that happen simultaneously within a **single forward pass of the original baseline model** (e.g., LLaDA).
>
> Therefore, the number of parameters is identical:
> - **Baseline (LLaDA):** 8 Billion parameters
> - **WINO (using LLaDA):** 8 Billion parameters

---

> ### Author Response · Authors · 2025-11-26
>
> Dear Reviewer Rjq2,
>
> Thank you for your time in reviewing our paper. We are writing to kindly mention that we have posted a detailed response to your questions on Nov 22.
>
> In our response, we provided the additional latency comparisons you requested and clarified the distinction between our method and the standard re-masking baselines. Crucially, we also clarified that WINO is a training-free algorithm that uses the original model weights, introducing no new parameters.
>
> We would value your feedback on whether these clarifications resolve your concerns before the discussion period ends.
>
> Best regards,
>
> Authors of Submission 2256

---

### Official Review · Reviewer_hTSB · 2025-11-01

**Soundness:** 3
**Presentation:** 3
**Contribution:** 3
**Rating:** 6
**Confidence:** 4

**Summary:**

The paper proposes Wide-In, Narrow-Out (WINO), a decoding method for accelerating inference in DLLMs. WINO appends a shadow block to the end of the context so that, after a single forward update, the model can write new tokens in parallel at several positions within the current block and also verify whether tokens written earlier in that block are reasonable, re-masking those deemed incorrect. Two hyperparameters, a drafting threshold and a verification threshold, allow a balance between throughput and accuracy. Because WINO is training-free and plug-and-play, it can be adopted by other DLLMs. The method is validated across multiple benchmarks and baselines.

**Strengths:**

1. The work introduces a novel draft-verify procedure that replaces the naive write-many-at-once update used in prior DLLM decoding. Compared with earlier approaches, WINO maintains high speedups without sacrificing performance and in many benchmarks even improves accuracy.
2. The study evaluates both text-only and vision-language settings over 14 tasks, which strengthens the credibility of the conclusions.
3. WINO does not require retraining and is easy to deploy as a plug-and-play component in other DLLMs.

**Weaknesses:**

1. As described in Section 4.2, during verification each position in y_shad conditions on the corresponding position in y_curr from the previous step. It is unclear whether this one-step lag negatively affects performance.
2. The paper lacks detailed hardware information for the experiments, especially which GPU models and how many were used to obtain the main TPS results.

**Questions:**

1. Why did the authors choose to append an additional auxiliary block at the end of the context to run the verification step on the freshly drafted block? Were alternative designs explored to achieve the same goal, for example adding trainable layers or parameters inside the model?
2. The draft-verify workflow can be interpreted as explicit self-refinement during DLLM generation, where the decision to re-mask a position is the refinement step. Under this view, allowing more reflection steps during inference could trade speed for higher accuracy. However, in the threshold-tuning ablations (Fig. 4), accuracy often rises at first and then declines as the number of steps increases. Do the authors have a plausible explanation for this pattern?

---

> ### Author Response · Authors · 2025-11-21
> **Response to Reviewer hTSB (1/2)**
>
> > As described in Section 4.2, during verification each position in y_shad conditions on the corresponding position in y_curr from the previous step. It is unclear whether this one-step lag negatively affects performance.
>
> Thank you for this question, which gets to the very core of WINO's design.
>
> Your understanding is **exactly correct**. There is indeed a one-step lag by design: tokens that are drafted in step $k$ are only eligible for verification in step $k+1$.
>
> This design is a deliberate and fundamental trade-off to ensure **computational efficiency**.
>
> 1.  **Why the Lag is Necessary:** The core premise of WINO is to perform *both* drafting new tokens and verifying existing tokens within a **single forward pass**.
>     * To verify a token $v$ at position $l$ (which was drafted at step $k-1$), we need to see if the model, given the *full context* from step $k$, would still generate $v$. This is what our shadow block does.
>     * To verify a token *just drafted* at step $k$, we would need to run an *entirely new forward pass* (step $k+1$) to build a new context that *includes* that brand-new token.
>     * Verifying a token in the *same step* it is drafted (i.e., a "zero-step lag") would require two forward passes per iteration (one to draft, a second to verify), effectively doubling the computation and completely negating the acceleration.
>
> 2.  **Impact on Performance:** Your question is whether this lag *negatively* affects performance. Our empirical results strongly indicate that it does not. In fact, this iterative refinement is the very mechanism that *improves* performance.
>
>     * **Case Study (Fig. 5 & Appendix A):** Our case studies explicitly demonstrate this "lag-and-correct" dynamic in action. In the GSM8K example, an incorrect token is drafted (e.g., "96" at step 37 or "86" at step 42). Because of the one-step lag, this error isn't caught immediately. However, in a *subsequent* step, the verification module identifies this low-quality token, re-masks it, and the model refines it to the correct answer ("84" at step 51).
>     * **Ablation Study (Table 4):** This is further proven by our ablation study. The "Only Draft" variant, which drafts aggressively *without any verification*, performs significantly worse than WINO. This shows that our lagged, one-step-behind verification is *crucial* and highly effective at correcting the errors introduced by aggressive drafting.
>
> In summary, the one-step lag is the key design choice that allows WINO to balance aggressive parallel drafting with iterative, high-quality refinement, all while maintaining a single forward pass per step. This mechanism is shown to be highly effective at mitigating error accumulation and is the primary driver of WINO's ability to improve accuracy while accelerating inference.
>
>
>
> > The paper lacks detailed hardware information for the experiments, especially which GPU models and how many were used to obtain the main TPS results.
>
> Thank you for flagging this omission. We apologize for the lack of clarity.
>
> All Tokens Per Second (TPS) and memory measurements reported in our main results  were benchmarked on a single NVIDIA A100 (80GB) GPU.
>
> We have added this specific hardware information to the "Implementation Details" subsection (5.1) to ensure full reproducibility.

---

> ### Author Response · Authors · 2025-11-21
> **Response to  Reviewer hTSB (2/2)**
>
> > Why did the authors choose to append an additional auxiliary block at the end of the context to run the verification step on the freshly drafted block? Were alternative designs explored to achieve the same goal, for example adding trainable layers or parameters inside the model?
>
> This is a great question regarding our architectural choices.
>
> We chose the appended auxiliary block design for two primary reasons:
>
> - **Training-Free & Plug-and-Play:** Our core design objective was to create a method that works instantly on existing, pre-trained DLLMs (like LLaDA and MMaDA) without requiring any fine-tuning or additional parameters. The auxiliary block leverages the pre-trained model's own bidirectional attention capabilities to perform "self-verification" simply by manipulating the attention mask and position IDs.
>
> - **Single-Pass Efficiency:** By appending the shadow block to the input, we can execute both the **Drafting**(on $Y_{\text{curr}}$) and **Verification** (on $Y_{\text{shad}}$) modules simultaneously in a **single forward pass**3. This is critical for inference speed.
>
> Regarding alternatives: We effectively ruled out designs involving **trainable layers or parameters** because they would violate the "training-free" constraint, limiting the method's generalizability to current open-source checkpoints. However, we do acknowledge in our "Future Directions" that integrating revokable sampling directly into the training phase is a promising avenue for potentially even greater performance.
>
>
> > The draft-verify workflow can be interpreted as explicit self-refinement during DLLM generation, where the decision to re-mask a position is the refinement step. Under this view, allowing more reflection steps during inference could trade speed for higher accuracy. However, in the threshold-tuning ablations (Fig. 4), accuracy often rises at first and then declines as the number of steps increases. Do the authors have a plausible explanation for this pattern?
>
> Thanks for your comment on the underlying trade-off.
>
> Our explanation for this pattern is twofold:
>
> - The Rise: The initial accuracy increase is the core benefit of WINO. As we add more refinement steps (e.g., by using a slightly more cautious tau1 or a stricter tau2), the model gets more chances to correct early drafting errors, which significantly boosts performance. This is WINO's "self-refinement" in action.
>
>
> - The Decline: The subsequent decline you noted occurs when the thresholds become too restrictive. An excessively high verification threshold (tau2) essentially limits the model's ability to explore possibilities, as any token that is not perfectly confident is immediately re-masked. In extreme cases, this over-cautiousness can cause the decoding process to "collapse" toward a serial-like procedure, losing the benefits of parallel exploration without a proportional gain in quality, and potentially getting stuck in a sub-optimal refinement loop.

---

### Official Review · Reviewer_ommK · 2025-11-03

**Soundness:** 3
**Presentation:** 2
**Contribution:** 3
**Rating:** 6
**Confidence:** 3

**Summary:**

This paper identifies the "irreversibility" of standard decoding in dLLMs as a primary cause for their poor quality-speed trade-off. To address this, the authors propose a training-free decoding algorithm named WINO: Wide-In, Narrow-Out. WINO introduces a revokable generation process through a parallel draft-and-verify mechanism. At each step, it uses a lenient threshold to aggressively draft multiple tokens ("Wide-In") and, in the same forward pass, leverages an auxiliary "shadow block" to re-evaluate previously generated tokens. Tokens that fail a stricter verification check are re-masked for later refinement ("Narrow-Out"). The method is evaluated on the LLaDA and MMaDA models, where it is shown to significantly reduce the number of decoding steps while often improving task performance.


While I am not against accepting the paper, I have a some concerns regarding the empirical evaluation and a lack of detail on the practical overheads. My primary reservations is on baselines. The paper's central claim of superiority is substantially weakened by the lack of comparison to other sophisticated, recently proposed dLLM acceleration techniques(but maybe they are concurrent work? depending on the definition of ICLR on this term). The experiments primarily compare against "standard" (one token at a time) and "naive parallel" decoding, which are weak baselines. Without benchmarking against other dynamic samplers, it is difficult to assess the true contribution of WINO over the state-of-the-art.

**Strengths:**

- **Clear and Intuitive Motivation:** The identification of "irreversibility" as a core problem is a strong conceptual contribution. The proposed draft-and-verify solution is a logical and elegant response to this problem. However, this seems to be aligning with many recent work, which might be worth including to related work, and the idea is also not super novel in AR LLMs community. Being honest with that will not hurt the novelty of the paper but establishes a great baseline.

- **Novel Mechanism:** The "shadow block" trick for enabling verification within a single forward pass is a clever and efficient design.

- **Promising Initial Results:** The ability to simultaneously reduce decoding steps by a large margin and improve accuracy on challenging benchmarks like GSM8K and ARC is a noteworthy and promising result. (but I have another concern that GSM8K seems to be a very contaminated dataset...)

- **Training-Free and Model-Agnostic:** The method's applicability to existing, pre-trained dLLMs without any fine-tuning is a major practical advantage.

**Weaknesses:**

- **Missing Key Baseline Comparisons:** This is the most significant weakness. The paper compares WINO to generating 1 token/step or a fixed M tokens/step. However, the related work section itself mentions other advanced samplers like Fast-dLLM-parallel and the entropy-bounded (EB) sampler, which also perform dynamic, confidence-based parallel decoding. To convincingly demonstrate the superiority of the WINO mechanism (specifically, the benefit of *revocation*), it is crucial to compare against these state-of-the-art dynamic samplers that perform aggressive drafting but lack a verification/revocation step. Without this, the reported gains might stem from simply being a better dynamic sampler, rather than from the novel revocation mechanism.

- **Lack of Analysis on Computational and Memory Overhead:**
     -  **Latency:** The speedup is reported in terms of step reduction and TPS. However, WINO increases the sequence length by adding a shadow block and requires a non-standard attention mask. This could slow down the per-step computation, especially if it prevents the use of highly optimized kernels like FlashAttention. A direct wall-clock time ablation comparing a forward pass with and without the WINO modifications (on sequences of the same effective length) is needed to understand the true per-step cost.

    - **Memory:** The paper claims the memory cost is "marginal." While likely true for the tested block length of 128, this cost is `(batch_size * Lb * d_model)`, which is non-trivial and scales with the block length `Lb`. A more rigorous analysis of how this memory overhead scales and its implications for longer-context decoding would strengthen the paper.

**Questions:**

- My most critical point is the comparison to other dynamic samplers. Can you provide results comparing WINO against Fast-dLLM-parallel and/or an entropy-bounded sampler on key benchmarks like GSM8K and Flickr30k? This would be essential to isolate the benefit of your novel verification/revocation mechanism.

- Could you provide a more detailed breakdown of the practical overheads? Specifically, what is the wall-clock time for a single forward pass with WINO compared to a standard forward pass on a sequence of equivalent length (i.e., `L_orig + Lb`)? This would clarify the true per-step latency cost of the custom attention mechanism.

- Additionally,  a formal pseudocode algorithm for the WINO decoding loop would be highly beneficial for clarity and reproducibility.

---

> ### Author Response · Authors · 2025-11-21
> **Response to Reviewer ommK (1/2)**
>
> > Missing Key Baseline Comparisons: This is the most significant weakness. The paper compares WINO to generating 1 token/step or a fixed M tokens/step. However, the related work section itself mentions other advanced samplers like Fast-dLLM-parallel and the entropy-bounded (EB) sampler, which also perform dynamic, confidence-based parallel decoding. To convincingly demonstrate the superiority of the WINO mechanism (specifically, the benefit of revocation), it is crucial to compare against these state-of-the-art dynamic samplers that perform aggressive drafting but lack a verification/revocation step. Without this, the reported gains might stem from simply being a better dynamic sampler, rather than from the novel revocation mechanism.
>
> We thank the reviewer for this critical comment, which targets the core of our contribution.
>
> You are correct that to prove the value of revocation, we must show that our "Draft" module alone is insufficient. We have two key pieces of evidence that address this.
>
> 1. This question is directly answered by our ablation study in **Table 4**.
>
> - The **"Only Draft"** variant is precisely the baseline you are describing: a strong, dynamic, confidence-based sampler (like Fast-dLLM-parallel) that aggressively drafts tokens but has **no verification or revocation** (i.e., $\tau_2=0$).
> - As Table 4 shows, this "Only Draft ($\tau_1=0.6$)" variant, while fast (7.36x speedup), suffers a **significant 3% accuracy drop** on GSM8K (70.28% vs. 73.24%).
> - Full **WINO**, by adding the verification/revocation module, not only recovers this 3% loss but *improves*accuracy by an additional 2.58%, achieving **75.82%**.
>
> This existing result already proves that the gains do *not* come from dynamic sampling alone. The **revocation mechanism** is the essential component that corrects errors and delivers the final high-quality performance.
>
> 2. New Experiment: Comparison with Fast-dLLM-parallel and Entropy-Bounded (EB) Sampler
>
>
>
> To further solidify this point, we have run the additional experiment you suggested. We compared WINO against the **Fast-dLLM-parallel** and **Entropy-Bounded (EB) sampler** on GSM8K and MMMU-val.
>
> | **GSM8K**          | **Accuracy** | **Steps** | **Step Reduction** | **TPS**    | **TPS Speedup** |
> | ------------------ | ------------ | --------- | ------------------ | ---------- | --------------- |
> | LLaDA (Baseline)   | 73.24        | 256.00    | 1.00x              | 17.76      | 1.00x           |
> | Fast-dLLM-parallel | 72.33        | 81.39     | 3.15x              | 56.12      | 3.16x           |
> | EB sampler  | 70.37        | 95.74     | 2.76x              | 46.71     | 2.63x           |
> | **WINO (Ours)**    | **75.82**    | **41.93** | **6.10x**          | **100.53** | **5.66x**       |
>
> | **MMMU-val**          | **Accuracy** | **Steps** | **Step Reduction** | **TPS**    | **TPS Speedup** |
> | ------------------ | ------------ | --------- | ------------------ | ---------- | --------------- |
> | MMaDA (Baseline)   | 18.56        | 256.00    | 1.00x              | 6.02      | 1.00x           |
> | Fast-dLLM-parallel | 19.89        | 35.63     | 7.18x              | 43.22      | 7.18x           |
> | EB sampler  | 17.87        | 61.04     | 4.20x              |24.89     | 4.13x           |
> | **WINO (Ours)**    | **24.00**    | **38.47** | **6.65x**          | **36.13** | **6.00x**       |
>
> This new comparison confirms that even advanced dynamic samplers, when pushed to aggressive speeds, suffer from the same severe performance degradation as naive methods. They lack the crucial ability to revoke and correct early errors.
>
>
> We have included this discussion into the revision.

---

> ### Author Response · Authors · 2025-11-21
> **Response to Reviewer ommK (2/2)**
>
> > Latency: The speedup is reported in terms of step reduction and TPS. However, WINO increases the sequence length by adding a shadow block and requires a non-standard attention mask. This could slow down the per-step computation, especially if it prevents the use of highly optimized kernels like FlashAttention. A direct wall-clock time ablation comparing a forward pass with and without the WINO modifications (on sequences of the same effective length) is needed to understand the true per-step cost.
>
> 1. TPS Already Includes All Overheads: The Tokens Per Second (TPS) speedup (e.g., 5.66x on GSM8K ) is an end-to-end, wall-clock time metric. It is calculated as (Total Tokens Generated / Total Wall-Clock Time). Note that the token number here refers to the actual number of generated tokens, excluding the shadow block. This metric already bakes in all overheads, including:
>     - The extra compute from the shadow block.
>     - The latency of applying the custom attention mask.
>     - Any other compute costs.
>
> 2. New Experiment: Per-Step Wall-Clock Ablation: To confirm this, we ran the exact ablation you suggested. We measured the wall-clock time for a single forward pass on the LLaDA-8B model.
>
> | **Configuration** | **Prompt Length** |**Block Length** | **Gen Length** |**Total Length** | **Attention Mechanism** | **Avg. Latency (ms)** |
> | ------ | ------| -----| ----- | ----- | --------- | ------- |
> | **Standard Pass** | 512                |   128       | 512            |    1024       | Standard Full Attention |        121.359 $\pm$ 0.367       |
> | **Standard Pass** | 512                  |  128      |  512+128            |   1024+128        | Standard Full Attention |   132.651  $\pm$ 0.600        |
> | **WINO Pass**     | 512                |  128        |  512            |  1024+128         | WINO Custom Mask        |     132.492 $\pm$ 0.889          |
>
>
> As shown, the cost of our custom attention mechanism is negligible.
>
> And we would like to clarify that modern attention kernels (e.g., FlashAttention) fully support **arbitrary binary attention masks without additional overhead**.
> In principle, one could implement Flash-style sparse attention to further reduce the FLOPs. However, given that the WINO mask exhibits only limited sparsity, such an optimization would yield negligible practical benefits.
>
>
> > Memory: The paper claims the memory cost is "marginal." While likely true for the tested block length of 128, this cost is (batch_size * Lb * d_model), which is non-trivial and scales with the block length Lb. A more rigorous analysis of how this memory overhead scales and its implications for longer-context decoding would strengthen the paper.
>
> Thank you for this insightful comment. We agree that a more detailed analysis of the memory overhead, especially as it scales, is beneficial. We acknowledge the reviewer's correct observation that the memory overhead from the auxiliary shadow block scales linearly with the block length $L_b$.
>
> - Our "marginal" claim was based on the semi-autoregressive setting ($L_b=128$) used in our main experiments. In this setting, the overhead is indeed minimal.
>
> - In addition, we would like to emphasize that the block lengths in existing decoding schemes are relatively small, and our experiments therefore reflect the actual memory consumption in practice.
>
> - To explicitly address the reviewer's concern about longer-context scaling, we ran a new experiment in the most memory-intensive "extreme" setting: **full diffusion decoding where generation_length = block_length ($L_b$) = 512**. This is the upper bound of WINO's overhead, as the auxiliary shadow block is as large as the entire output, effectively doubling the sequence length processed by the model (from 512 to 1024 tokens). Even in this case, the GPU memory overhead of our method remains below 8%.
>
> | **Method** | **Setting**        | **Peak GPU Memory** (GB) |
> | --- | ---- | --- |
> | MMaDA   | Gen Len = Block Len = 512 |  20.96            |
> | WINO  | Gen Len = Block Len = 512 |  22.65          |
>
>
>
> > Additionally, a formal pseudocode algorithm for the WINO decoding loop would be highly beneficial for clarity and reproducibility.
>
> Thank you for this excellent and constructive suggestion. In the revised manuscript, we have added a formal algorithm in Appendix B. This algorithm explicitly details:
>
> 1. The initialization of the current block $Y_{\text{curr}}$ and the auxiliary shadow block $Y_{\text{shad}}$.
> 2. The iterative refinement loop that continues until no [MASK] tokens remain in $Y_{\text{curr}}$.
> 3. The parallel **"Wide-In" (Drafting)** step, where multiple tokens are unmasked in $Y_{\text{curr}}$ based on the lenient threshold $\tau_1$.
> 4. The parallel **"Narrow-Out" (Verification)** step, which uses the shadow block's outputs and the stricter threshold $\tau_2$ to re-mask any low-confidence tokens.
>
> We appreciate the reviewer pointing this out.

---

### Official Review · Reviewer_qPHN · 2025-11-04

**Soundness:** 2
**Presentation:** 3
**Contribution:** 3
**Rating:** 6
**Confidence:** 3

**Summary:**

This paper introduces Wide-In, Narrow-Out (WINO), a novel training-free decoding algorithm for Diffusion Large Language Models (DLLMs) that enables revokable decoding. WINO's core contribution is a parallel draft-and-verify mechanism. It first aggressively generates a large set of candidate tokens, and then uses the model's bidirectional context to verify all currently unmasked tokens (including historical ones), re-masking any that are identified as low-quality. The experiments show that WINO improves the quality-speed trade-off, reducing inference steps while improving accuracy.

**Strengths:**

1. The proposed method is training-free, allowing it to function as a plug-and-play module for existing pre-trained DLLMs.
2. The algorithm demonstrates strong performance gains while accelerating the inference.

**Weaknesses:**

1. The proposed solution to prevent information leakage during verification seems insufficient. The paper states that a token in the Y_shad cannot attend to its corresponding token in Y_cur. However, it can attend to other tokens within Y_cur. Since full self-attention is applied within Y_cur, these other tokens already contain contextual information about the token being verified. Therefore, information leakage still exists. It causes doubt on the mechanism's claimed effectiveness, as the model is not re-evaluating the token with truly independent context.

2. The applicability of WINO to the full class of diffusion language models is unclear. It is not obvious how this method would apply to or benefit other diffusion frameworks, such as SEDD[1], which utilize different sampling schemes (e.g., τ-leaping) that are inherently iterative and corrective.

3. The decoding algorithm appears to be deterministic. The paper does not address how WINO could be adapted for stochastic sampling to produce diverse outputs for a single prompt.

[1] Lou et al., Discrete Diffusion Modeling by Estimating the Ratios of the Data Distribution.

**Questions:**

None

---

> ### Author Response · Authors · 2025-11-21
> **Response to Reviewer qPHN (1/2)**
>
> > The proposed solution to prevent information leakage during verification seems insufficient. The paper states that a token in the Y_shad cannot attend to its corresponding token in Y_cur. However, it can attend to other tokens within Y_cur. Since full self-attention is applied within Y_cur, these other tokens already contain contextual information about the token being verified. Therefore, information leakage still exists. It causes doubt on the mechanism's claimed effectiveness, as the model is not re-evaluating the token with truly independent context.
>
>
> That is an insightful observation. We sincerely thank the reviewer for this deep engagement with our proposed mechanism.
>
> You are theoretically correct that an indirect information path $Y_{curr}[l] \to Y_{curr}[m] \to Y_{shad}[l]$ exists, given that $Y_{shad}[l]$ attends to $Y_{curr}[m]$ and $Y_{curr}$ employs full self-attention.
>
> However, we argue that this theoretical, multi-hop path does not lead to practical information leakage that would compromise the verification mechanism. Our reasoning is as follows:
>
>
> - The Dominance of Direct Paths:
>     - Regarding the information flow or leakage from $Y_{\text{curr}}[l]$ to $Y_{\text{shad}}[l]$, there are two possible pathways: a **direct path** $Y_{\text{curr}}[l] \to Y_{\text{shad}}[l]$, and an **indirect path** $Y_{\text{curr}}[l] \to Y_{\text{curr}}[m] \to Y_{\text{shad}}[l]$.
>     - During training, the optimization process driven by gradient descent inherently favors the **simplest and most direct routes** for credit assignment. That is, the model primarily learns to exchange information through direct attention between the two positions. The model is neither trained nor encouraged to "read" the token at $Y_{\text{curr}}[l]$ by interpreting its weak "reflection" embedded within $Y_{\text{curr}}[m]$.
>     - Our method operates in a **training-free inference** setting. Once the dominant paths are severed, the model in practice lacks the capability to leak information through these indirect routes.
>
> - Empirical Ablation: To empirically prove that our mask is both necessary and effective, we conduct a new ablation study on the GSM8K benchmark. We compare our full WINO method against a new variant: "WINO (w/ Full Leakage)". In this new variant, $Y_{shad}[l]$ is allowed to directly attend to its corresponding token $Y_{curr}[l]$.
>     - When direct information leakage is permitted, the "WINO (w/ Full Leakage)" variant fails. This experiment confirms that the direct information path is the only one that practically matters, and our proposed attention mask  is both necessary and sufficient to block it, thereby enabling an effective verification and refinement process.
>
> We have added this crucial ablation study and the accompanying discussion to the revised manuscript (Tab. 4) to clarify the effectiveness of our mechanism.
>
>
> | **Method**                 | **Accuracy** | **Steps** | **Step Reduction** | **TPS**    | **TPS Speedup** |
> | -------------------------- | ------------ | --------- | ------------------ | ---------- | --------------- |
> | LLaDA (Baseline)           | 73.24        | 256.00    | 1.00x              | 17.76      | 1.00x           |
> | Only Draft ($\tau_1=0.6$)  | 70.28        | 34.79     | 7.36x              | 130.89     | 7.37x           |
> | **WINO (Ours)**            | **75.82**    | **41.93** | **6.10x**          | **100.53** | **5.66x**       |
> | **WINO (w/ Full Leakage)** |   72.25           |    44.85       |       5.71$\times$             |      92.38      |      5.20$\times$           |
>
> | **MMMU-val**                 | **Accuracy** | **Steps** | **Step Reduction** | **TPS**    | **TPS Speedup** |
> | -------------------------- | ------------ | --------- | ------------------ | ---------- | --------------- |
> | MMaDA (Baseline)   | 18.56        | 256.00    | 1.00x              | 6.02      | 1.00x           |
> | **WINO (Ours)**    | **24.00**    | **38.47** | **6.65x**          | **36.13** | **6.00x**       |
> | **WINO (w/ Full Leakage)** |   18.22           |    44.43       |       5.76$\times$             |      31.54      |      5.24$\times$           |

---

> ### Author Response · Authors · 2025-11-21
> **Response to Reviewer qPHN (2/2)**
>
> > The applicability of WINO to the full class of diffusion language models is unclear. It is not obvious how this method would apply to or benefit other diffusion frameworks, such as SEDD[1], which utilize different sampling schemes (e.g., τ-leaping) that are inherently iterative and corrective.
>
> We appreciate the opportunity to clarify WINO's scope. WINO is specifically tailored for Masked Diffusion Models (MDMs) (e.g., LLaDA, MMaDA, MDLM), which represent the current state-of-the-art and most scalable paradigm in diffusion LLMs.
>
> The distinction is fundamental and mathematical:
> 1. SEDD relies on a specialized ratio-based score parameterization to enable correction. This method is mathematically incompatible with MDMs.
> 2. MDMs achieve superior scalability and performance by directly modeling $p_\theta(x_0|x_t)$, but standard decoding in this framework is inherently "absorbing" and irreversible.
>
> WINO is the first to unlock "revokable" capabilities for the mainstream MDM framework. It introduces a novel mechanism to solve the critical irreversibility bottleneck in SOTA models, effectively marrying the high performance of MDMs with the corrective benefits previously limited to frameworks like SEDD, all without retraining.
>
> > The decoding algorithm appears to be deterministic. The paper does not address how WINO could be adapted for stochastic sampling to produce diverse outputs for a single prompt.
>
> We thank the reviewer for this observation. The reviewer is correct that our algorithm, as presented in the main paper using arg max for drafting, is deterministic.
>
> This is not a fundamental limitation. WINO can be easily adapted for stochastic sampling.
>
> Instead of the deterministic arg max, we can introduce Gumbel noise to the model's output logits. This standard Gumbel-Max technique introduces stochasticity, allowing for diverse outputs on repeated runs. The verification module remains unchanged, as its role is to deterministically ensure the quality and coherence of the (now stochastically drafted) tokens.
>
>
> We report temperature = 0.5 in the below table. We have clarified this and added the new experiments in the revised manuscript (Tab. 5).
>
>
>
> | GSM8K | Acc (%)  | Step reduction ($\times$) |
> | ----- | ---- | -------------- |
> | LLaDA | 73.01 $\pm$ 0.21 | -           |
> | WINO      | 76.06 $\pm$ 0.16     |      6.05 $\pm$ 0.06           |
>
> | MMMU-val | Acc (%)  | Step reduction ($\times$) |
> | ----- | ---- | -------------- |
> | MMaDA | 18.32 $\pm$ 0.43 | -           |
> | WINO      | 23.89 $\pm$ 0.29     |      6.69 $\pm$ 0.09           |

---

> > ### Comment · Reviewer_qPHN · 2025-11-27
> >
> > Thank you for the detailed response, which resolve my concerns.

---

> > > ### Author Response · Authors · 2025-11-27
> > >
> > > Thank you for your follow-up and for confirming that your concerns have been addressed. We appreciate your time and constructive feedback.

---

### Author Response · Authors · 2025-11-21
**General Response**

We strictly appreciate the time and effort all reviewers dedicated to evaluating our work. We are encouraged that the reviewers unanimously recognized the novelty and practical value of **WINO**.

1. Common Ground & Highlights. We are glad that the reviewers reached a consensus on the core contributions of this paper:

- **Innovative & Intuitive Concept**: Reviewers praised the identification of "irreversibility" as a **fundamental bottleneck** in Diffusion LLMs and the "**logical and elegant**" (ommK) solution of **revokable decoding**. The draft-and-verify mechanism was highlighted as a "**novel**" (hTSB, ommK) and "**core contribution**" (qPHN) that effectively solves the problem.
- **Practicality & Ease of Use**: All reviewers (qPHN, ommK, hTSB, Rjq2) commended the **training-free**and **plug-and-play** nature of WINO, recognizing its ability to be seamlessly applied to existing pre-trained DLLMs without the need for expensive retraining.
- **Superior Performance:** The method's ability to decisively improve the **quality-speed trade-off** was acknowledged by all reviewers. Specifically, the **significant speedups** (e.g., 6x-10x) combined with **accuracy improvements** on challenging benchmarks were described as "**strong**" (qPH), "**promising**" (ommK), and "**decisive**" (Rjq2).

2. Summary of Revisions

In response to the constructive feedback, we have extensively revised the manuscript (changes marked in **teal** text) and conducted comprehensive new experiments to further solidify our claims:

- **Comparisons with SOTA Dynamic Samplers:** To isolate the specific benefit of our revocation mechanism, we added comparisons against advanced dynamic samplers, including **Fast-dLLM-parallel** and the **Entropy-Bounded (EB) sampler** (suggested by ommK). The results (added in **Appendix C**) confirm that WINO significantly outperforms these non-revokable baselines.
- **Rigorous Validation of Mechanism:** We added a crucial ablation study, "WINO (w/ Full Leakage)," to empirically prove that our custom attention mask is both necessary and sufficient to prevent information leakage during verification, addressing the theoretical concern raised by qPHN.
- **Detailed Efficiency & Overhead Analysis:** We included a direct measurement of **per-step wall-clock latency**, confirming that our custom attention mechanism incurs negligible overhead compared to standard attention. We also added a **memory** scaling analysis under extreme conditions (full diffusion with long contexts) to address concerns from ommK and Rjq2.
- **Expanded Versatility & Reproducibility:** We demonstrated that WINO supports **stochastic sampling**(via Gumbel-Max) for diverse generation (qPHN). Additionally, we added a formal **pseudocode algorithm**(in **Appendix B**) to enhance clarity and reproducibility (ommK).

We believe these revisions have significantly strengthened the paper, firmly establishing WINO as a robust, efficient, and high-quality decoding standard for Diffusion LLMs. We have provided detailed point-by-point responses to each reviewer below.

Best regards,

Authors of Submission 2256

---

### Meta-Review · Area_Chair_etiz · 2025-12-27

**Summary:**

This paper proposes a novel training-free decoding algorithm for Diffusion Large Language Models (DLLMs) that enables revokable decoding. The method is easy to use and provides flexibility. The primary concerns reviewers have are: 1) the absence of baselines; 2) misrepresentation of the evaluation metrics. During the rebuttal, the authors have added the full comparison with baselines and provided additional explanation of the evaluation details. After considering all the discussions, I recommend acceptance, but I strongly suggest that the authors revise the manuscript in light of the reviewers' feedback.

**Reviewer Concerns:**

1) missing some baseline methods
2) The evaluation is not well organized.

**Reviewer Scores:**

Based on the rebuttal, I think Reviewer Rjq2 will increase the overall rating from 2 to 4. For other reviewers, I think they would stay unchanged.

---

### Decision · Program_Chairs · 2026-01-26

Accept (Poster)